# Incorporation of Suppression of Tumorigenicity 2 into Random Survival Forests for Enhancing Prediction of Short-Term Prognosis in Community-ACQUIRED Pneumonia

**DOI:** 10.3390/jcm11206015

**Published:** 2022-10-12

**Authors:** Teng Zhang, Yifeng Zeng, Runpei Lin, Mingshan Xue, Mingtao Liu, Yusi Li, Yingjie Zhen, Ning Li, Wenhan Cao, Sixiao Wu, Huiqing Zhu, Qi Zhao, Baoqing Sun

**Affiliations:** 1Cancer Centre, Institute of Translational Medicine, Faculty of Health Sciences, University of Macau, Macau 999078, China; 2MoE Frontiers Science Center for Precision Oncology, University of Macau, Macau 999078, China; 3Department of Allergy and Clinical Immunology, Department of Laboratory, National Center for Respiratory Medicine, National Clinical Research Center for Respiratory Disease, State Key Laboratory of Respiratory Disease, Guangzhou Institute of Respiratory Health, The First Affiliated Hospital of Guangzhou Medical University, Guangzhou 510120, China

**Keywords:** community-acquired pneumonia, suppression of tumorigenicity 2, random survival forests, clinical stability

## Abstract

(1) Background: Biomarker and model development can help physicians adjust the management of patients with community-acquired pneumonia (CAP) by screening for inpatients with a low probability of cure early in their admission; (2) Methods: We conducted a 30-day cohort study of newly admitted adult CAP patients over 20 years of age. Prognosis models to predict the short-term prognosis were developed using random survival forest (RSF) method; (3) Results: A total of 247 adult CAP patients were studied and 208 (84.21%) of them reached clinical stability within 30 days. The soluble form of suppression of tumorigenicity-2 (sST2) was an independent predictor of clinical stability and the addition of sST2 to the prognosis model could improve the performance of the prognosis model. The C-index of the RSF model for predicting clinical stability was 0.8342 (95% CI, 0.8086–0.8598), which is higher than 0.7181 (95% CI, 0.6933–0.7429) of CURB 65 score, 0.8025 (95% CI, 0.7776–8274) of PSI score, and 0.8214 (95% CI, 0.8080–0.8348) of cox regression. In addition, the RSF model was associated with adverse clinical events during hospitalization, ICU admissions, and short-term mortality; (4) Conclusions: The RSF model by incorporating sST2 was more accurate than traditional methods in assessing the short-term prognosis of CAP patients.

## 1. Introduction

Community-acquired pneumonia (CAP) is a common and deadly infectious disease that kills 3 million people worldwide each year and places a significant strain on global health care systems [1]. Despite continuous improvements and breakthroughs in medical conditions in recent decades, CAP remains a significant cause of ICU admission and death in adults [2,3,4]. The current treatment for most CAP patients is antibiotic therapy, but studies have shown that approximately 6% to 15% of hospitalized patients do not achieve clinical improvement after early treatment [5,6]. Patients with clinical improvement have a mortality rate several times lower than that of patients with clinical failure [7,8]. Therefore, it is an urgent need for some accurate markers and models that can screen inpatients with a low probability of cure early in their admission and can help doctors to adjust the management of CAP patients and improve patient prognosis.

To date, the clinical application of index-based prognosis remains very limited, with the CURB-65 score and the Pneumonia Severity Index (PSI) score remaining the most commonly used indicators to determine the prognosis of CAP patients [9,10,11]. However, several studies indicate that these two scores do not perform well in predicting the risk of ICU admission and death, so new severity scores are needed to predict the prognosis of CAP patients [12,13,14,15]. Over time, machine learning based methods are nowadays increasingly being used to predict CAP patients [16,17,18,19]. For example, machine learning models using the random forest algorithm have been successfully applied to predict 30-day mortality in pneumonia patients with higher accuracy than the CURB-65 score [16]. Inspired by those cases, here we hypothesize that this newly developed method randomized survival forest (RSF) can be utilized to predict the clinical stability of CAP patients at 30 days.

In addition, the development of some new biological markers can further improve the prognosis of patients [17,18,19]. Suppression of Tumorigenicity-2 (ST2) is a member of Interleukin (IL)-1 superfamily, and 2 major transcription variants have been identified: the full length of transmembrane form (ST2L) and the soluble form (sST2). sST2 is a decoy receptor for IL-33 [20]. It can competitively bind IL-33, thus blocking the function of the IL-33-ST2L signaling pathway, which is an important cardioprotective paracrine system [21,22]. As early as the 2013 ACC/AHA Heart Failure Guidelines, sST2 was recommended for predicting the probability of in-hospital and long-term death in patients with heart failure, and its role as a predictive biomarker of cardiac death has been well documented [23,24]. In addition to heart failure, the potential use of sST2 in lung diseases is also of interest as it is significantly elevated in patients with idiopathic pulmonary fibrosis, asthma, and pulmonary hypertension [25,26]. sST2 has been found to be a predictor of all-cause in-hospital mortality of CAP patients [27]. In this study, the prognostic value of sST2 in CAP patients was investigated and evaluated. More importantly, we aim to develop and validate the RSF model incorporating sST2 to assess patients’ in-hospital prognosis, assessing its prognostic performance in a cohort study and comparing it with CURB-65 and PSI scores.

## 2. Materials and Methods

### 2.1. Experimental Design

A total of 247 CAP patients who were hospitalized at the First Hospital of Guangzhou Medical University in 2019–2020 were included in this study. Their clinical information was collected from their electronic medical records with a 30-day in-hospital follow-up. The diagnostic criteria of patients were current recommendations for the treatment of community-acquired pneumonia [28]. Patients were diagnosed with CAP when new infiltrative shadows appeared on chest radiograph and one of the following acute respiratory signs and symptoms were present: cough, sputum production, dyspnea, the core body temperature of 38 °C or higher, abnormal respiratory response or gong sounds on auscultation, and white blood cell count higher than 10 × 10^9^/L or less than 4 × 10^9^/L. The exclusion criteria were (1) age < 20 years; (2) lack of clinical information to assess PSI score and CURB-65 score; (3) patients with severe immunodeficiency: infection with human immunodeficiency virus or CD^4^+ cells less than 350 cells/μL; (4) prolonged hospitalization, hospitalization 90 days before, and wound care 30 days before because it suggested the diagnosis of medical-associated pneumonia; and (5) refusal to sign an informed consent form or to follow the physician’s consultation.

### 2.2. Clinical Information and Outcomes

A trained research assistant extracted and collected chest radiograph reports, demographic (age, sex), clinical (temperature, pulse, respiratory rate, systolic blood pressure, state of consciousness, ventilator support or not and complications) and laboratory information (Procalcitonin, C-reactive protein, blood counts, coagulation markers, biochemical markers, and blood gas analysis) from the hospital electronic case system and did PSI and CURB-65 scores. When the chest radiograph report of pneumonia was ambiguous, we sought the opinion of our radiologist.

To assess patients’ ability to recover, we defined the primary clinical outcome event as clinical stability. We judged the clinical stability of patients according to the current CAP guidelines of the American Thoracic Society (ATS) and the study of Claudine Angela Blum et al. [28,29]. Clinical stability was defined when the patient met all of the following signs: a temperature ≤ 37.8 °C, heart rate ≤ 100 beats/min, spontaneous respiratory rate ≤ 24 breaths/min, and systolic blood pressure ≥ 90 mmhg without pressor intervention (≥100 mmhg in hypertensive patients), normal mental status, spontaneous oral intake, and partial pressure of oxygen in artery (PaO2) ≥ 60 mm Hg or pulse oximetry ≥ 90%. The number of days to achieve clinical stability was recorded as the time from the patient’s admission to clinical stability for at least 24 h. In addition, we defined ICU admission, 30-days in-hospital all-cause death, and in-hospital adverse clinical events as secondary clinical outcome events (including septicaemia, sepsis, shock, respiratory failure, empyema, ARDS, ventilator support, and heart disease).

### 2.3. Detection of sST2

Venous blood was collected within the first day (0–24 h) after the patients were admitted to the hospital. Whole blood was centrifuged at 3000× *g* for 15 min at room temperature. Serum was extracted and stored at −80 °C. The serum samples were prepared by the sST2 detection kit (ST22101001F) and then the sST2 levels were detected by the Jet-iStar 3000 (Joinstar Biomedical Technology Co., Ltd., Hangzhou, China), according to the manufacturer’s instruction. The primary antibody in the kit is an anti-sst2 mouse monoclonal antibody, and the secondary antibody is a CFSE-labeled anti-sst2 mouse monoclonal antibody. The absorbance of the carboxyfluorescein diacetate n-succinimidyl ester (cfse)-labeled anti-sST2 mouse monoclonal antibody-sst2-anti-sst2 mouse monoclonal antibody complex was measured at 635 mm. Finally, quantitative sST2 levels were obtained from a standard curve drawn from standard concentration solutions measured on the same plate.

### 2.4. Data Analysis

Continuous data were expressed as mean (±standard deviation) and the difference was assessed by student’s *t*-test. Categorical variables were expressed as frequency (percentage) and the difference was assessed by chi-square test. The missing values were imputed by multivariate imputation via chained equations [30]. Variables were selected in two steps. First, correlation analysis (Pearson correlation coefficient) and variance analysis were applied for the initial screening of continuous and categorical variables, respectively. Variables that were not correlated with clinical stability were removed. Next, the least absolute shrinkage and selection operator (LASSO) algorithm was performed to further select and sort significant variables. LASSO algorithm minimizes the regression coefficients through a continuous shrinkage algorithm to reduce the possibility of overfitting. The R package “glmnet” was used in our study to implement the LASSO regression.

Our prognostic models were developed using random survival forest (RSF) with patients reaching clinical stability as the primary outcome. RSF is a nonparametric machine learning method that is an extension of Breiman’s random forest for analyzing survival data. The RSF model is trained by growing a large number of individual trees. Since it does not require restrictive parameters or proportional survival assumptions, the RSF model is more applicable than some traditional survival models. The method was implemented in the R package “randomForestSRC”, and the RSF model was trained using 500 trees and a terminal node size of 15 [31].

We evaluated the performance of our models in future populations through internal bootstrap validation. The training set was obtained by repeated sampling among patients using the bootstrap method, and the test set is all patients. Internal discrimination of our models was assessed by the time-dependence receiver operating characteristic (time-dependent ROC) curve and concordance index (C-index). The area under the curve (AUC) was calculated and compared. We further applied the RSF model to the entire dataset and calculated the RSF score for each patient. The X-title was used to find the optimal threshold to classify patients into high-risk and low-risk subgroups. The log-rank test was used to compare the Kaplan–Meier survival curves for each subgroup. The significant difference was set to 0.05. Analyses were performed in the R language environment (version 4.0.2, R Foundation for Statistical Computing, Vienna, Austria). 

## 3. Results

### 3.1. Patient Characteristics

The basic information of the 247 eligible CAP patients is shown in Table 1. In total, 208 of them (84.21%) reached clinical stability within 30 days of follow-up. The age of patients who reached clinical stability was significantly younger than those who did not (*p* < 0.05), and fewer of the patients who reached clinical stability had a pulse >125 beats/min, respiratory rate > 30 breaths/min, and systolic blood pressure (SBP) < 90 mmhg (*p* < 0.001). In addition, D-dimer, Procalcitonin (PCT), C-reactive protein (CRP), and sST2 were significantly lower in patients who reached clinical stability than in those who did not (*p* < 0.05), while Lymphocyte and Eosinophils were higher in patients who reached clinical stability (*p* < 0.001).

### 3.2. Variable Selection

A total of 60 indicators were collected from patients, including demographic information, clinical assessment, and laboratory tests. After removing those with high missing values (more than 5%) and redundant variables, 39 variables were left for follow-up analysis. A total of 11 continuous indicators were selected as the variables correlated with clinical stability using a correlation network analysis (Appendix A). For the categorical variables, patients presenting with pleural effusion, mental confusion, SBP < 90 mmhg and respirations > 30 breaths/min took more time to reach clinical stability (*p* < 0.001) (Appendix A). The LASSO regression was conducted in these 15 variables to reduce overfitting. The model performs best when the number of variables was 10 (Appendix A), namely, Age, D-dimer, sST2, Neutrophils, Lymphocyte, Prothrombin activity (PTA), glucose, Blood urea nitrogen (BUN), SBP < 90 mmhg and respirations > 30 breaths/min. Univariate regression analysis proved that all indicators were significant predictors of clinical stability (*p* < 0.05) (Table 2). Lymphocyte and PTA are favorable factors for clinical stability (HR > 1), that is, patients with high Lymphocyte and PTA were more likely to reach clinical stability, while other predictors were unfavorable factors for clinical stability (HR < 1).

### 3.3. Prognostic Value of sST2

The potential use of sST2 in different lung diseases is also gaining attention. Multivariate regression analysis found that sST2 was an independent predictor of clinical stability (*p* < 0.001). We compare the sST2 values in patients with different prognoses, such as clinical stability, ICU admission, and survival status. The significant differences of sST2 in patients with different prognoses were found (Figure 1A). The ST2 could predict clinical stability, ICU admission, and 30-day mortality, with AUCs of 0.8826 (95% CI: 0.8345–0.9306), 0.9218 (95% CI: 0.8837–0.96), and 0.9018 (95% CI: 0.8402–0.9634), respectively (Figure 1B). The optimal threshold for sST2 was found to be 28.2 ng/mL using X-title and patients were divided into high- and low-sST2 groups. The Kaplan–Meier curves for patients to reach clinical stability were compared, and log-rank showed a significant difference between the high- and low-sST2 groups (Figure 1C). Patients with low-sST2 took less time to reach clinical stability than patient with high-sST2, and 97.4% (38 of 39 patients) with high-sST2 did not reach clinical stability.

### 3.4. Model Development and Comparison

The cox regression model and RSF models to predict the clinical stability of CAP patients were developed based on the selected 10 variables. To validate the additive effect of sST2 in the model, we developed and compared two models, i.e., the RSF model without sST2 and the RSF model with sST2. The models were trained using the training set obtained from bootstrap, and then the entire dataset was used as a test set to evaluate the performance of the model. The performance metrics of CURB 65, PSI score, cox regression model, and two RSF models were compared (Table 3) with the C-index of 0.7181, 0.8025, 0.8214, 0.8336, and 0.8595, respectively. The 10-days AUC, 20-days AUC, and 30-days AUC of RSF score were 0.9740 (95% CI, 0.9574–0.9907), 0.9573 (95% CI, 0.9284–0.9863), and 0.9428 (95% CI, 0.9113–0.9744), respectively. The AUC and C-index of the two RSF models were higher than those of CURB 65, PSI score and cox regression model, and the C-index of the RSF model with sST2 was higher than that of the RSF model without sST2. 

The RSF scores were calculated for each patient in the entire data set to evaluate the predictive performance of the RSF model. The optimal threshold of RSF score was found to be 0.297 using X-title, so the patients could divide into the high-score (RSF score ≥ 0.297) group and low-score (RSF score < 0.297) group. The Kaplan–Meier curves for those two groups showed significant differences assessed by log-rank test (*p* < 0.001) (Figure 2A). Patients with low-score took less time to reach clinical stability than those with high score (Figure 2B). The RSF score performed well in predicting clinical stability of patients, with the 30-days AUC as high as 0.9428 (95% CI:0.9113–0.9744) (Figure 2C). In addition, the RSF scores were positively correlated with PA-aO2 and CRP (Figure 2D) (R = 0.5808 and 0.6649, *p* < 0.001), and RSF scores were significantly higher in patients with higher CURB 65 and PSI scores (Figure 2E,F). 

### 3.5. Potential of RSF Score to Predict Adverse Clinical Events

We also documented several adverse clinical events that occurred during the patient’s hospitalization, such as septicemia, sepsis, shock, respiratory failure, empyema, ARDS, ventilator support, and heart disease. The results show that patients who experienced these adverse events had higher RSF scores than those who did not (*p* < 0.05) (Figure 3A). It suggests that patients with high RSF scores are more prone to adverse clinical events. For patients in the low-score group, no patient died within 30 days of follow-up, and no patient was admitted to the intensive care unit (ICU) within 30 days of hospitalization (Figure 3B). The AUCs of sST2, CURB-65, PSI score, and RSF score in predicting adverse clinical events were evaluated and compared (Table 4). The AUCs of RSF scores appears to be higher than others except for septicemia, where PSI has a higher performance.

## 4. Discussion

Our study demonstrated that sST2 was an independent predictor of clinical stability by cox regression analysis and sST2 could predict clinical stability, with AUCs of 0.8826 (95% CI: 0.8345–0.9306). Further, an RSF model was derived and internally validated to evaluate the prognostic risk of adult CAP patients by cleverly embedding sST2 into routine. This model can accurately assess the probability to achieve clinical stability of adult CAP patients during hospitalization with a C-index of 0.8595 (0.8445–0.8745). With the incorporating of sST2, the ability of the RSF model to stratify risk and assess prognosis in CAP patients was significantly higher than that of CURB-65 and PSI scores. In addition, the RSF model had potential in assessing the risk of adverse clinical events, ICU admissions, and short-term mortality during hospitalization. 

Our study illuminated that patients with a younger age, lower D-dimer, lower sST2, 270 lower Neutrophils, lower glucose, higher PTA, higher Lymphocyte, no SBP <90 mmHg 271, and no breathing rate >30 breaths/min were more likely to achieve clinical stability. Previous studies have confirmed that cardiovascular complications remain a heavy burden of poor course and prognosis in CAP patients in which coagulation factors play an important role [32]. It also has been found that D-dimer levels were significantly elevated in patients with clinical failure or severe CAP [33]. Our findings are consistent with them that low levels of D-dimer and high levels of PTA are favorable factors for reaching in-hospital clinical stability in CAP patients. The possible mechanism is that acute infection in CAP leads to an increase in myocardial metabolic demand, while pre-existing heart-related diseases make myocardial ischemia, which would prolong hospital stay and increase mortality in patients [34]. In addition, the endotoxin of the Gram-negative pathogen that triggers CAP would make an imbalance in the coagulation-fibrinolytic system in the body, resulting in abnormal levels of coagulation factors, such as D-dimer [35]. Inflammatory cells are also important factors affecting the prognosis of CAP patients. The neutrophil-to-lymphocyte ratio (NLR) is a simple, rapid, and inexpensive biomarker of systemic inflammation. As reported, it has been widely used for prognosis in different pathological conditions (pathology) [36]. This further confirms that the model proposed in this study adequately considers the impact of the host systemic inflammatory response in CAP patients on their prognosis and can be better adapted to the clinical reality. The model also demonstrated that admission hyperglycemia is an unfavorable factor for patients to reach clinical stability, consistent with the fact that admission fasting hyperglycemia is an independent risk factor for complications and death in CAP patients [37]. This may be due to the acute stress response of the body to infection.

Consistent with previous study [26], sST2 still had a good predictive performance for CAP patient prognosis in cohort studies with larger numbers of patients and more clinical variables. IL-33-ST2L plays an important regulatory role in pulmonary infections. In a mouse model of sepsis, administration of IL-33 treatment improved inflammation and reduced mortality [38]. In addition, in a mouse model of COPD exacerbation caused by an influenza virus infection, IL-33 treatment increased neutrophil infiltration in the lung, whereas sST2 treatment decreased this infiltration [39]. The expression of IL-33 is high in bronchial epithelial cells and pulmonary vascular endothelial cells. When the body is under damage and stimulation, IL-33 rapidly positively release from damaged cells or is secreted by immune cells [40,41]. Among a variety of inflammatory respiratory diseases, including CAP, injured cells secreted and released IL-33. IL-33 bound to ST2L in immunocyte and activated ST2L/IL-1RacP, recruiting MyD88 to its intracellular domain. MyD88 binding recruits IL-1R-associated kinase (IRAK-1 and IRAK-4) and TRAF6, leading to the NF-κB pathway being activated. NF-κB activations induce pro-inflammatory gene transcription and promote inflammatory cytokine expressions [42,43]. Lin et al., reported that, in human corneal epithelial cells, IL-33—ST2L signaling was enhanced after exposure to IL-33, which promoted the expression of proinflammatory cytokines and chemokines in both mRNA and protein levels [44,45]. IL-33 and proinflammatory cytokines can induce chemotactic migration of neutrophils to the lung and enhance the pulmonary inflammatory response [46]. However, the NF-κB signaling pathway not only induced inflammation but promoted the expression of sST2 [47]. When IL-33-ST2L promotes inflammation and kills pathogens, sST2 may be secreted in increased amounts in lung tissues and competitively bound to IL-33 due to a negative feedback mechanism, blocking the IL-33-ST2L signaling pathway (Figure 4). Therefore, sST2 is expected to serve as a biomarker of poor prognosis in CAP patients.

The PSI is a scoring tool jointly developed by the American Thoracic Society (ATS) and Infectious Diseases Society of America (IDSA) that can predict short-term mortality in patients with CAP by classifying them into five classes. Due to its accuracy, rigor, validity, and safety, the PSI score has become the reference standard for CAP risk stratification [48]. However, the practical value of PSI in clinical practice has been controversial due to the number of variables required (including demographics, clinical characteristics, laboratory data, and chest radiographs) and the long calculation time. In contrast, the CURB-65 score requires only five clinical variables and is much simpler to calculate. In practice, the CURB-65 score is easier to administer than the PSI, but it is less sensitive in predicting mortality in CAP patients. In addition, the specificity of both scores is weak [49]. Several studies have demonstrated that the prognostic performance of machine learning models was better than PSI and CURB-65 [9,12,13,16,50,51,52]. In our study, the 30-day AUC and C-index of the RSF model were 0.9428 (95% CI, 0.9113–0.9744) and 0.8595 (95% CI, 0.8445–0.8745), respectively, both higher than those of PSI score and the CURB-65 score. The inflammatory response has been recognized as a key aspect of prognosis in CAP patients [53]. Importantly, our RSF model includes factors, such as neutrophil count, lymphocyte count, and sST2, that reflect systemic or local inflammatory response in CAP patients, whereas neither PSI score nor CURB-65 score can assess host inflammatory response. This may be responsible for the apparent superiority of our new model over PSI and CURB-65 scores.

Our model divided the patients into the high-score and low-score groups based on their normalized RSF scores. The optimal threshold was found to be 0.297 using X-title. When the patient’s RSF score was less than the threshold, the patient was able to reach clinical stability rapidly in a short period. Additionally, according to the analysis of secondary outcomes, patients in the low-risk group were found to be at little risk of ICU admission and death in the short term. When the patient’s RSF score was greater than the threshold, it took longer for the patient to reach clinical stabilization and there was a possibility of ICU admission and death. In addition, the higher the patient’s RSF score, the higher the likelihood of adverse clinical events.

Our study still has several limitations. We only included samples from the First Affiliated Hospital of Guangzhou Medical University. Although the Department of Respiratory Medicine of the First Affiliated Hospital of Guangzhou Medical University enjoys a national reputation, external validation of the new cohort is needed. Second, although previous studies and our study have demonstrated that is sST2 can be used as an early monitoring indicator for CAP patients, sST2 is not yet a routine screening test in most hospital. The cost of sST2 testing is not affordable for all patients and hospitals. In addition, only short-term hospitalization information was available for this study, and further studies need to be implemented to collect long-term follow-up information on patients to re-validate our results. Finally, the construction of the model is complex and requires a large amount of data for validation. There is still a lot of work to be done to apply the model to daily practice.

## 5. Conclusions

We prospectively developed a well-calibrated RSF model that can evaluate the prognostic risk of adult CAP patients and predict whether the patients can reach clinical stability. The RSF model in prognostic evaluation seems to have a better performance than pre-existing CURB 65 and PSI score. After further validation and modification, this model could assist clinical management decisions to improve the care of hospitalized CAP patients.

## Figures and Tables

**Figure 1 jcm-11-06015-f001:**
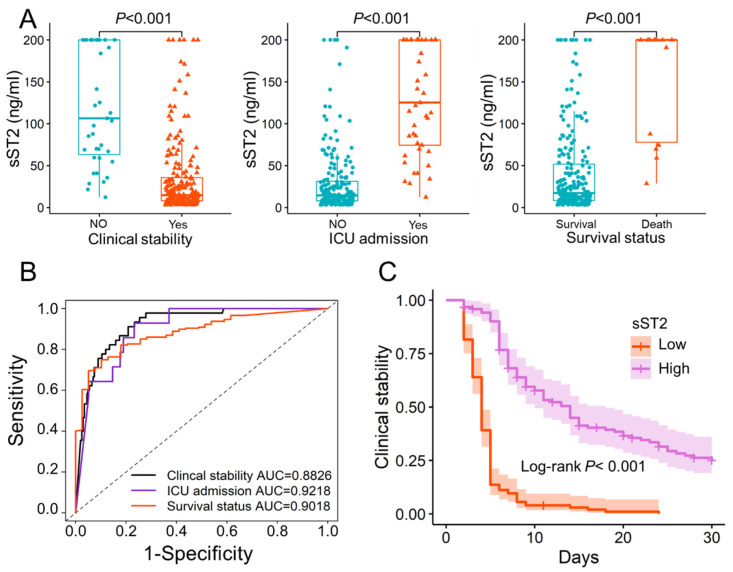
The potential prognostic value of sST2. (**A**). The significant differences of sST2 in patients with different prognoses. The sST2 of patients who reached clinical stability (34.20 ± 45.84) was significantly lower than that of patients who did not (120.24 ± 68.37) (*p* < 0.001). sST2 was significantly higher in patients admitted to the ICU (129.59 ± 64.43) than in patients not admitted to the ICU (29.54 ± 38.90) (*p* < 0.001). Survival patients (150.72 ± 68.36) had significantly higher sST2 than patients who died in hospital (41.63 ± 52.45) (*p* < 0.001). (**B**). The ROC curves of sST2 in predicting clinical stability, ICU admission and 30 days in-hospital mortality. (**C**). The Kaplan–Meier curve of patients with low-sST2 (sST2 < 28.2 ng/mL) and high-sST2 (sST2 > 28.2 ng/mL). The optimal threshold of sST2 was 28.2 ng/mL obtained with X-title. sST2: Soluble Form of Suppression of Tumorigenicity-2; ICU: Intensive care unit; Low: low-sST2; High: high-sST2; AUC: Area under the curve.

**Figure 2 jcm-11-06015-f002:**
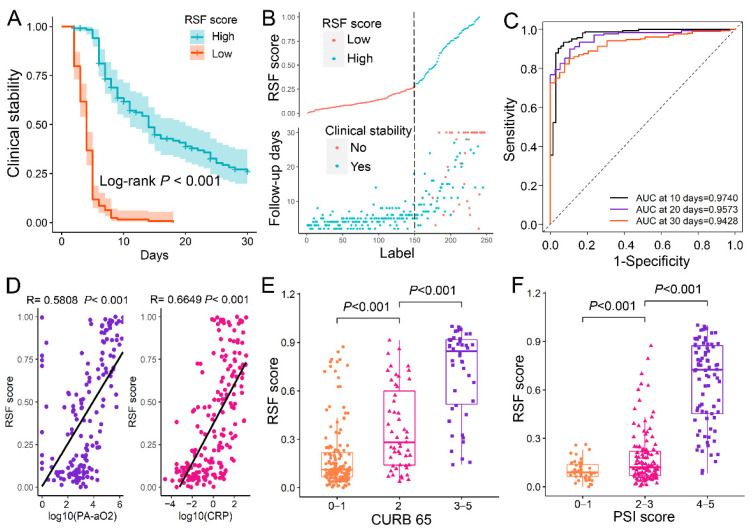
The prognostic value of RSF model. (**A**). The Kaplan–Meier curve of clinical stability for patients with different RSF scores. The patients were divided into the high-score (≥0.297) and low-score (<0.297) based on their RSF score, and the optimal threshold was 0.297 obtained with X-title. (**B**). The correlation between RSF score and the time to clinical stability. (**C**). The time-dependence ROC curve of RSF scores. (**D**). The RSF scores were positively correlated with the PA-aO2 and CRP. (**E**). Patients with high CURB-65 scores (3–5 points) had higher RSF scores than patients with intermediate CURB-65 scores (2 points) and further higher than those with low CURB-65 scores (0–1 points) (*p* < 0.001). (**F**). Patients with high PSI scores (grades 3–5) had higher RSF scores than patients with intermediate PSI scores (grades 2–3) and further higher than those with low PSI scores (grades 0–1) (*p* < 0.001). RSF: Random survival forest, CPR: C-reactive protein; PA-aO2: Alveolar-arterial PO2 difference; CURB 65: Confusion, Urea level, Respiratory rate, Blood pressure, and Age > 65 years; PSI: Pneumonia Severity Index; Low: RSF scores < 0.297; High: RSF score ≥ 0.297; AUC: Area under the curve.

**Figure 3 jcm-11-06015-f003:**
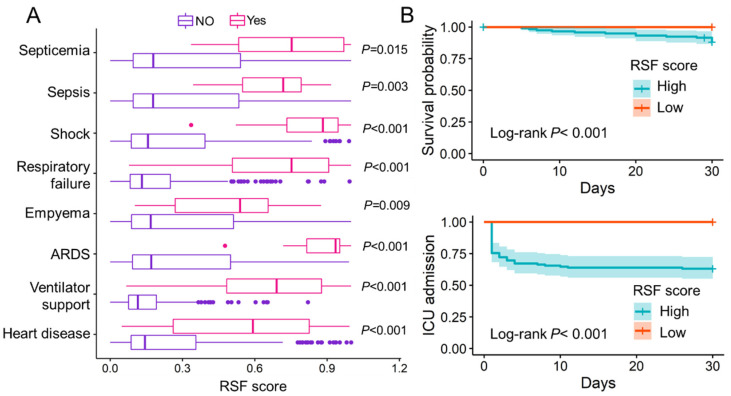
The evaluation of RSF score to predict adverse clinical events. (**A**). Patients who experienced the following adverse clinical events, including septicemia, sepsis, shock, respiratory failure, empyema, ARDS, ventilator support and heart disease, had significantly higher RSF scores than patients who did not experience these adverse clinical events (*p* < 0.05). (**B**). Kaplan–Meier curve of in-hospital death and ICU admission for patients in different risk groups. ARDS: Acute respiratory distress syndrome; RSF: Random survival forest; ICU: Intensive care unit; Low: RSF scores < 0.297; High: RSF score ≥ 0.297.

**Figure 4 jcm-11-06015-f004:**
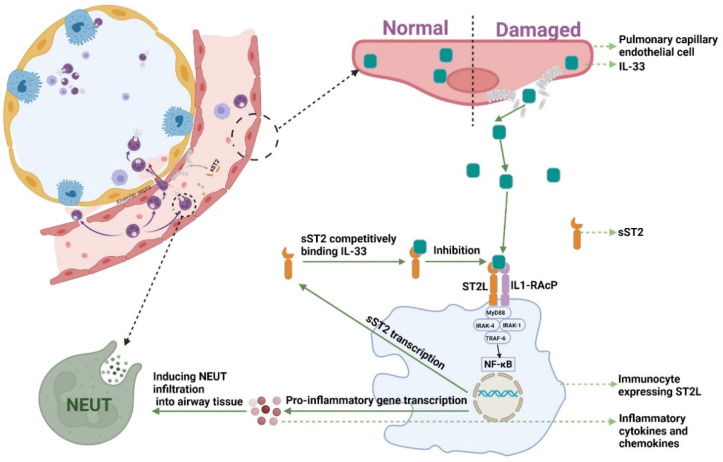
The mechanism of sST2-IL-33 signaling pathway. NF-κB: Nuclear Factor-κ-Gene Binding; MyD88: myeloid differentiation factor 88; IRAK-1: interleukin-1 receptor associated kinase 1; IRAK-4: interleukin-4 receptor associated kinase 1; TRAF6: TNF receptor associated factor 6; NEUT: neutrophils; IL-33: Interleukin-33; sST2: Soluble Form of Suppression of Tumorigenicity-2; sST2L: Full-length Transmembrane Form Suppression of Tumorigenicity-2.

**Table 1 jcm-11-06015-t001:** Basic information of the CAP patients. Continuous variables were expressed as mean (±standard deviation) and categorical variables were expressed as frequency (percentage).

Variables	Clinical Stability		*p*
	No (*n* = 39)	Yes (*n* = 208)	
Age	66.38 (±12.91)	57.46 (±16.74)	0.002
Male	24 (61.54)	123(59.13)	0.918
*Clinical assessment*			
Temperature ≤ 35 °C or ≥40 °C	39 (100.0)	208 (100.0)	
Pulse ≥ 125 beats/min	5 (12.8)	8 (3.8)	0.056
Breaths > 30 breaths/min	29 (74.4)	71 (34.1)	<0.001
SBP < 90 mmhg	6 (15.4)	5 (2.4)	0.001
Ventilator support	38 (97.4)	48 (23.2)	<0.001
State of consciousness	17 (43.6)	8 (3.8)	<0.001
*Laboratory tests*			
HBP (ng/mL)	76.83 (±84.72)	41.58 (±49.43)	<0.001
PCT (ng/mL)	1.94 (±3.46)	0.63 (±2.39)	0.004
sST2 (ng/mL)	120.24 (±68.37)	34.20 (±45.84)	<0.001
CRP (ng/mL)	6.86 (±5.72)	2.62 (±4.42)	<0.001
Leukocyte (×0^9^/L)	11.04 (±5.43)	8.33 (±3.23)	<0.001
Neutrophils (×10^9^/L)	9.53 (±4.85)	5.83 (±3.30)	<0.001
Lymphocyte (×10^9^/L)	0.79 (±0.57)	1.61 (±0.83)	<0.001
Monocyte (×10^9^/L)	0.63 (±0.46)	0.66 (±0.28)	0.599
Eosinophils (×10^9^/L)	0.75 (±1.13)	2.67 (±3.04)	<0.001
Basophils (×10^9^/L)	0.25 (±0.26)	0.53 (±0.38)	<0.001
Platelets (×10^9^/L)	221.74 (±142.89)	247.56 (±103.57)	0.186
Hematocrit (×10^9^/L)	0.28 (±0.07)	1.16 (±11.39)	0.633
Total bilirubin (μmol)	23.28 (±33.43)	12.59 (±9.79)	<0.001
Direct bilirubin (μmol)	10.24 (±19.57)	3.11 (±5.03)	<0.001
D-dimer (ng/mL)	3584.75 (±3354.47)	1008.30 (±1673.25)	<0.001
Prothrombin time (s)	16.50 (±5.10)	13.97 (±1.59)	<0.001
Prothrombin activity (%)	76.56 (±22.28)	93.31 (±16.39)	<0.001
Fibrinogen (g/L)	3.87 (±1.64)	4.13 (±1.37)	0.306
APTT (s)	44.82 (±13.14)	39.91 (±6.13)	<0.001
FiO2 (%)	48.46 (±19.64)	27.31 (±27.70)	<0.001
PaCO2 (mmHg)	48.05 (±10.70)	44.47 (±7.70)	0.022
SpO2 (%)	95.75 (±5.24)	96.49 (±4.00)	0.356
PA-aO2 (mm Hg)	206.02 (±141.19)	44.67 (±61.59)	<0.001
PaO2 (mm Hg)	100.54 (±35.87)	100.06 (±31.81)	0.936
Creatinine (μmol/L)	92.03 (±45.66)	75.84 (±22.40)	0.001
PH	7.39 (±0.07)	7.38 (±0.04)	0.57
Na(mmol/L)	144.67 (±6.89)	139.84 (±4.29)	<0.001
Glucose(mmol/L)	8.45 (±3.74)	5.96 (±2.36)	<0.001
BUN (mmol/L)	13.29 (±8.56)	6.40 (±4.83)	<0.001
PSI score (%)			<0.001
1	0 (0.0)	42 (20.2)	0.010
2	1 (2.6)	69 (33.2)	0.002
3	2 (5.1)	48 (23.1)	0.030
4	16 (41.0)	41 (19.7)	0.080
5	20 (51.3)	8 (3.8)	<0.001
CURB 65 score (%)			<0.001
0	0 (0.0)	70 (33.7)	<0.001
1	6 (15.4)	74 (35.6)	0.060
2	8 (20.5)	43 (20.7)	1.000
3	16 (41.0)	19 (9.1)	<0.001
4	6 (15.4)	2 (1.0)	<0.001
5	3 (7.7)	0 (0.0)	0.003
*Complications and adverse clinical events*			
Heart disease (%)	17 (43.6)	38 (18.3)	0.001
ARDS (%)	12 (30.8)	1 (0.5)	<0.001
Empyema (%)	5 (12.8)	13 (6.2)	0.266
Respiratory failure (%)	30 (76.9)	35 (16.8)	<0.001
Septicemia (%)	4 (10.3)	2 (1.0)	0.004
Shock (%)	18 (46.2)	8 (3.8)	<0.001
Pleural effusion (%)	18 (46.2)	58 (27.9)	0.038
Sepsis (%)	2 (5.1)	5 (2.4)	0.678
Death (%)	17 (43.6)	0 (0.0)	<0.001
ICU (%)	27 (69.2)	18 (8.7)	<0.001

CAP: Community-acquired pneumonia; SBP: Systolic blood pressure; HBP: Heparin-Binding Protein; PCT: Procalcitonin; CRP: C-reactive protein; APTT: Activated partial thromboplastin time; BUN: Blood urea nitrogen; ARDS: Acute respiratory distress syndrome; ICU: Intensive care unit; PSI: Pneumonia Severity Index.

**Table 2 jcm-11-06015-t002:** Cox regression analysis of the selected variables.

Variables	Univariate Regression		Multivariate Regression	
	HR (95% CI)	*p*	HR (95% CI)	*p*
Age	0.9830 (0.9753–0.9908)	<0.001	0.9934 (0.9831–1.0039)	0.216
D-dimer	0.9997 (0.9996–0.9998)	<0.001	0.9998 (0.9997–1.0000)	0.012
sST2	0.9853 (0.9913–0.9874)	<0.001	0.9905 (0.9853–0.9956)	<0.001
Neutrophils	0.8734 (0.8371–0.9113)	<0.001	0.9400 (0.8898–0.9930)	0.027
Lymphocyte	1.6889 (1.4820–1.9247)	<0.001	1.1254 (0.8871–1.4277)	0.331
PTA	1.0322 (1.0242–1.0403)	<0.001	1.0171 (1.0065–1.0279)	0.002
Glucose	0.8237 (0.7715–0.8794)	<0.001	0.9665 (0.8995–1.0358)	0.352
BUN	0.8777 (0.8415–0.9154)	<0.001	0.9974 (0.9605–1.0358)	0.893
SBP	0.2931 (0.1206–0.7126)	0.007	0.7072 (0.2728–1.8331)	0.475
Breaths	0.3843 (0.2867–0.5151)	<0.001	0.5340 (0.3776–0.7751)	<0.001

HR: Hazard ratio; sST2: Soluble Form of Suppression of Tumorigenicity-2; PTA: Prothrombin activity; BUN: Blood urea nitrogen; SBP: Systolic blood pressure.

**Table 3 jcm-11-06015-t003:** Performance metrics of CURB 65, PSI score, cox regression model, and two RSF models in test set.

Model	AUC			C-Index
	10 Days	20 Days	30 Days	
Curb 65	0.8438(0.7914–0.8961)	0.8454 (0.7823–0.9085)	0.8553 (0.8028–0.9172)	0.7181(0.6933–0.7429)
PSI Score	0.9212(0.8853–0.9585)	0.8989(0.8545–0.9433)	0.8810(0.8554–0.9441)	0.8025(0.7776–0.8274)
Cox regression	0.9379(0.9044–0.9714)	0.9179(0.8718–0.9640)	0.8900(0.8421–0.9380)	0.8214(0.8080–0.8348)
RSF without sST2	0.9653(0.9485–0.9822)	0.9558(0.9274–0.9842)	0.9450(0.9141–0.9760)	0.8336(0.8214–0.8458)
RSF with sST2	0.9740(0.9574–0.9907)	0.9573(0.9284–0.9863)	0.9428(0.9113–0.9744)	0.8595(0.8445–0.8745)

CURB 65: Confusion, Urea level, Respiratory rate, Blood pressure, and Age > 65 years; PSI: Pneumonia Severity Index; sST2: Soluble Form of Suppression of Tumorigenicity-2; AUC: Area under curve.

**Table 4 jcm-11-06015-t004:** The AUCs of ST2, CURB-65, PSI score and RSF score in predicting adverse clinical events of patients.

	sST2	CURB 65	PSI Score	RSF Score
Septicemia	0.7576(0.6074–0.9078)	0.7894(0.6434–0.9354)	0.8292(0.7037–0.9546)	0.8154(0.6705–0.9602)
Sepsis	0.7711(0.6817–0.8605)	0.6247(0.4491–0.8003)	0.7726(0.6825–0.8627)	0.8006(0.7004–0.9008)
Shock	0.8708(0.8221–0.9195)	0.8307(0.7493–0.9120)	0.8807(0.8328–0.9286)	0.9210(0.8782–0.9637)
Respiratory failure	0.8561(0.7992–0.9131)	0.8009(0.7438–0.8580)	0.8737(0.8315–0.9159)	0.8990(0.8574–0.9406)
Empyema	0.6417(0.5238–0.7596)	0.5528(0.4118–0.6937)	0.6887(0.5711–0.8063)	0.7086(0.6188–0.7985)
ARDS	0.8490(0.7568–0.9411)	0.8665(0.7608–0.9723)	0.9121(0.8657–0.9584)	0.9287(0.8699–0.9874)
Ventilator support	0.8876(0.8396–0.9356)	0.8552(0.8119–0.8985)	0.9185(0.8838–0.9532)	0.9222(0.8863–0.9581)
Heart disease	0.7411(0.6639–0.8183)	0.7644(0.7006–0.8282)	0.8049(0.7475–0.8622)	0.7721(0.7017–0.8425)
ICU admission	0.9018(0.8637–0.9400)	0.8506(0.7952–0.9060)	0.8958(0.8574–0.9341)	0.9516(0.9255–0.9777)
Death	0.8818(0.8202–0.9434)	0.8758(0.7811–0.9706)	0.9109(0.8713–0.9504)	0.9207(0.8691–0.9722)

AUC: area under curve; CURB 65: Confusion, Urea level, Respiratory rate, Blood pressure, and Age > 65 years; PSI: Pneumonia Severity Index; RSF: Random survival forest; sST2: Soluble Form of Suppression of Tumorigenicity-2; ARDS: Acute respiratory distress syndrome; ICU: Intensive care unit.

## Data Availability

The data that support the findings of this study are available from the corresponding authors.

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
