# Peer review of "Incorporation of Suppression of Tumorigenicity 2 into Random Survival Forests for Enhancing Prediction of Short-Term Prognosis in Community-ACQUIRED Pneumonia"

_jcm, 2022, doi:10.3390/jcm11206015_

Round 1
Reviewer 1 Report
Authors looked at sST2 as a biomarker for CAP with RSF to enhance accuracy than the traditional methods.
1. It has been noted in the literature that with disease severity, IL-33 levels go up. Since sST2 is a decoy receptor for IL-33, did authors see a correlative change in IL-33 levels in these patients? Adding IL-33 levels in the paper will strengthen the data.
2. Authors mention why they think sST2 may be upregulated in CAP patients. They also suggest that IL-33 may be therapeutic. However, it is not clear how and why sST2 and IL-33, both go up in several respiratory diseases, including CAP.
3. A cartoon suggesting the mechanism will be helpful for readers to comprehend what happens with disease and homeostasis.
4. More relevant and recent publications in the field of IL-33 and ST2 should be included, particularly in respiratory disease.
Author Response
We would like to thank the respected reviewer for his useful comments. We have tried to consider all comments and revised the manuscript based on the comments. Please see the attachment for specific modifications.

Reviewer 2 Report
The development of a model predicting the outcomes of patients with community acquired pneumonia is dramatically important for the every-day activity of any Emergency Room and Pulmonology unit. For this reason, I think that well designed studies in this field, as the one the Authors have here submitted, are really important and welcome.
Here I report some issues regarding the submitted work that I consider important to be fixed.
Abstract
1. “Background: The development of biomarkers and models can screen inpatients with a low probability of cure in the early stages of admission to help doctors adjust the management of community-acquired pneumonia (CAP) patients”.
This sentence needs to be expressed in a more fluent fashion, as the other parts of abstract are written.
Introduction
2. In line 42 Authors cite reference 2 referring globally to the finding of that work, but it discusses a specific geographical area and its results don’t appear to be generalizable. I think that another reference should be added to make the sentence more appropriate: “Cavallazzi R, Furmanek S, Arnold FW, Beavin LA, Wunderink RG, Niederman MS, Ramirez JA. The Burden of Community-Acquired Pneumonia Requiring Admission to ICU in the United States. Chest. 2020 Sep;158(3):1008-1016. doi: 10.1016/j.chest.2020.03.051. Epub 2020 Apr 13. PMID: 32298730.”.
3. In line 45 Authors refers to reference 7, but it could be more appropriate to cite it in the part of introduction where the clinical applications of sST2 are described, maybe with a dedicate sentence.
4. Line 51, reference “Arnold FW, Brock GN, Peyrani P, Rodríguez EL, Díaz AA, Rossi P, Ramirez JA; CAPO authors. Predictive accuracy of the pneumonia severity index vs CRB-65 for time to clinical stability: results from the Community-Acquired Pneumonia Organization (CAPO) International Cohort Study. Respir Med. 2010 Nov;104(11):1736-43. doi: 10.1016/j.rmed.2010.05.022. Epub 2010 Jun 23. PMID: 20576417.” appears to be more suitable than reference number 9.
5. Line 54, correct “With the advent of time” with “over time”.
6. Line 58, correct “Inspired for those cases” with “Inspired by those cases”.
7. Line 72, merge citation 25 and 26.
Materials and Methods
8. In line 102-103 the Authors declare that “When the chest radiograph report of pneumonia was ambiguous, we sought the opinion of our respiratory medicine investigator”. What “respiratory medicine investigator” stands for? Was him/her a radiologist? An explanation is needed.
9. Line 115: correct “inncluding”.
10. in the Data analysis section, Authors need to explain the statistical methods used to analyze continuous and categorical values showed in Table 1. Furthermore, it is not explained the p value considered for statistically significance.
Results
10. In line 159 correct “The basic information of the 247 eligible CAP patients (Table 1)” with "The basic information of the 247 eligible CAP patients is showed in Table 1”.
11. Table 1 needs to be revised:
- for continuous variables, it is more appropriate to express them with mean (± standard deviation) or median (interquartile range) and indicate it in the caption, while for categorical values it is better to explain in the caption that number and percentages are expressed
-for PSI and CURB-65 scores, could be useful and more complete to insert also the p values for every subgroup
-enrich the caption with the explanation of every abbreviation used in the table.
12. Check for the use of the acronym “LASSO” throughout the text, because sometimes is written in capitals and others not.
13. Table 2: enrich the caption with the explanation of every abbreviation used in the table.
14. Line 187: correct “prognosis” with “prognostic”.
15. Line 189: correct “founded” with “found”.
16. Figure 1: add p values in the figures, where expected (A and C), explain in the text the mean or median value of sST2 and the p values for every category of analysis (clinical stability, ICU admission and survival status), add in the caption the explanation of the abbreviation used.
17. Line 220: correct “table 2” with “table 3”.
18. It is better to explain in the “Model Development and comparison” section how the high and low score of RSF model has been obtained other than in the discussion section, where it is currently described.
19. Figure 2: add the name of the abscissa axis in figure 2B, add p values in the figures where expected (E and F) and explain in the text the mean or median value of CURB-65 and PSI and the p values for every category of analysis in the figure; explanation of the abbreviation in caption is also required.
20. The section “Potential of RSF Score to predict adverse clinical events” needs to be better explained. The Authors declared that “The results show that the occurrence of these 245 events will significantly increase the RSF score of patients”: is the RSF score a predictor of complication or is it affected by the occurrence of complication? This is not clear.
21. Figure 3: explain the features of figures 3A and add p values in figures 3A and 3B.
22. Line 250-251: add that AUC of RSF appears to be higher except for septicemia, where PSI has a higher performance (as showed in table 4).
Discussion
23. The Authors declared that “Our study illuminated that patient with younger age, lower D-dimer, lower sST2, 270 lower Neutrophils, lower glucose, higher PTA, higher Lymphocyte, no SBP <90mmHg 271 and no breathing rate >30 breaths/min had a poor short-term prognosis”: is it a typo? Because the data showed that these features are associated with clinical stability.
24. Line 319: correct “weaker” with “weak”.
25. In line 339-340 the Authors declare “Although the First Hospital of Guangzhou Medical University has the best respiratory department in China”. Please tone down this affirmation.
26. In line 350 the Authors write “The performance of the model in prognostic evaluation is better than”: please tone down this sentence (i.e., seem to have a better performance, or similar).
27. Although sST2 seems to be a promising biomarker, it is not available and easily applicable in all situations and centers, as rightly stated by the Authors. Also, the economic aspect has to be pointed out in the discussion, maybe within the limitations.
28. Regarding the limitations, another important point is the complexity of the model the Authors built, because it appears to be difficult to apply in the every-day practice (as PSI is) due to the great amount of data needed. Point out this issue in the discussion.
Supplementary material
29. Add indication of p value in figure S1B.
Author Response

(The authors gave the same response as above.)

Round 2
Reviewer 1 Report
Authors have made substantial changes and improved the article.